# Association between Emotion Dysregulation and Distinct Groups of Non-Suicidal Self-Injury in Taiwanese Female Adolescents

**DOI:** 10.3390/ijerph16183361

**Published:** 2019-09-11

**Authors:** Wan-Lan Chen, Chin-Cha Chun

**Affiliations:** 1Department of Human Development and Psychology, Tzu Chi University, Hualien 97074, Taiwan; 2Psychiatric Department, Buddhist Tzu Chi General Hospital, Adjunct Clinical Psychologist, Hualien 97004, Taiwan; 3Department, Nurture- mind Psychological Clinic, New Taipei City 23441, Taiwan; cardigan220@yahoo.com.tw

**Keywords:** emotion dysregulation, female adolescents, negative affect, non-suicidal self-injury

## Abstract

*Background*: Previous studies revealed that female adolescents are more likely than males to engage in non-suicidal self-injury (NSSI) to regulate negative emotions; however, the dimensions of emotion regulation that are associated with NSSI behavior in adolescents require further examination. The present study aimed to identify Taiwanese female adolescent clusters with NSSI engagement frequency and to evaluate the association of specific forms of emotion dysregulation with NSSI. *Methods*: The participants were 438 female adolescents (mean age = 15.23 years, SD = 1.24, range between 13 and 18) recruited from 11 high schools. Self-report questionnaires assessing NSSI, difficulties in emotion regulation, and positive and negative affect were administered, and 37% of respondents reported a history of NSSI. *Results*: The analysis of NSSI frequency yielded three groups: severe, moderate, and non-NSSI. High negative affect, low positive affect, and difficulties in all aspects of emotion regulation differentiated female adolescents in the severe NSSI group from their counterparts in the non-NSSI group. The moderate and severe NSSI groups were further distinguished by age of onset, negative affect, emotion regulation strategies, and impulse control. Adolescents classified in the severe group reported earlier onset of NSSI, higher negative affect, less emotion regulation strategies, and more difficulty with impulse control. *Conclusions:* The results indicate that assessments of NSSI and emotion regulation should be incorporated in youth mental health screening. The clinical implications of NSSI behavior intervention require further discussion.

## 1. Background

Adolescence is a crucial stage in human development during which critical biological, psychological, and social changes occur [1]. Some adolescents experience greater emotional ups and downs following these changes, and, if adequate psychological resources to deal with emotions are lacking [2], their mental health might be affected due to emotion dysregulation [3]. Among the mental health problems related to emotion regulation, non-suicidal self-injury (NSSI) drew the attention of scholars, practitioners, and educators in recent years. NSSI comprises behaviors in which people intentionally inflict damage to the surface of their body [4]. This type of injury differs from cultural rituals, involuntary injury resulting from driving under the influence of alcohol, or even repeated injurious behavior induced by cognitive or other pathological factors [5,6,7]. Although there is a clear distinction between NSSI companied by suicidal thoughts and suicidal behavior, studies found that NSSI is highly associated with, and may coexist with, suicide or suicidal attempts [8,9]. Therefore, parents, teachers, and mental health clinicians need to monitor the NSSI behavior of teenagers with greater vigilance. 

Previous research on NSSI behavior focused on populations in clinical settings [10], but more attention is now being paid to nonclinical populations [11,12,13]. The results of these studies showed that the onset of NSSI behavior mostly occurs between ages 12 and 14 [8,14,15,16]. Nearly one out of every three to four teenagers exhibits NSSI behavior at some stage. Although some studies indicated that there is no gender difference in the prevalence rate of NSSI [17], others showed that the rate of female adolescents engaging in NSSI is greater than that of male adolescents [18,19]. Previous studies even indicated that female adolescents are two [20,21] or even eight times [22] more likely to engage in NSSI behavior than men. The study of Nixon et al. [23] showed that girls account for 86% of adolescent inpatients who exhibit NSSI behavior. Sourander et al. [24] found that boys and girls show few differences in NSSI behavior at the initial stage of adolescence (approximately 12 years of age), while, by the middle stage of adolescence (approximately 15 years of age), gender differences in NSSI behavior gradually emerge, with far more female than male adolescents involved. Nock and Prinstein [25] found that NSSI usually occurs in the context of psychological distress, and adolescent girls usually have more psychological distress than men [22,23,26,27]. Therefore, it is necessary to pay more careful attention to NSSI by female adolescents. Past research also indicated a high prevalence of NSSI in Taiwan. Lin et al. [28] reported a prevalence of 20.1% among Taiwanese high-school students, and females reported significantly higher frequency of NSSI than did males. 

Self-injurious behavior by teenagers usually takes the form of cutting, while scratches, burns, or self-beating are also common [17,29,30]. Many self-injured people reported using multiple methods to injure themselves [6,31]. The frequency of NSSI behavior differs greatly among the self-injurious population; some may have made one or two experimental attempts, while others engage more frequently in NSSI behavior. Even within the same gender, heterogeneity may appear due to differences in the methods [32] or frequency [33,34] of NSSI. Given the widespread prevalence of NSSI, the exploration of these differences is an important priority. 

A number of empirical studies confirmed the hypothesis that NSSI behavior may serve a negative emotional regulation function for some individuals (e.g., References [35,36]). Emotion regulation encompasses different aspects of the emotion process [37] including awareness, understanding, and acceptance of emotion, and regulating strategies [38]. However, research on which emotion regulation dimension deficits are significantly relevant to NSSI is still limited. The present study aims to explore the relationship between NSSI behavior and difficulty with emotion regulation in Taiwanese female adolescents, in particular, to determine which dimensions of emotion regulation difficulty have distinct predictive effects on NSSI. Furthermore, this study discusses the characteristics of subgroups which may differ in the frequency of NSSI behavior and in the dimension of emotion regulation. This research is expected to provide valuable insight into the NSSI behavior of female adolescents in Taiwan for early identification of girls with self-injury behavior. 

## 2. NSSI Behavior and Emotion Regulation

According to Linehan’s [39] theory of emotion dysregulation in borderline personality disorder (BPD), individuals with BPD may experience more intense and longer-lasting negative emotions, and, since they have limited capacity to regulate emotion due to genetic characteristics, an early invalidating environment, or both, self-injury becomes a means of reducing intense negative emotions. Relevant research provided ample evidence for the applicability of Linehan’s model to NSSI [35,36,40,41,42,43]. Individuals often describe a sense of calmness as a result of the release of emotional tension after engaging in NSSI behavior [36,40] and, thus, NSSI serves as form of negative reinforcement [15], implying that individuals adopt NSSI as a coping strategy for regulating aversive emotional experiences. The positive correlation between the frequency of NSSI and the function of negative reinforcement was indicated by Klonsky [42].

Based on the theoretical work of Linehan, Gratz and Roemer [38] developed the Difficulties in Emotion Regulation Scale (DERS) to assess the association between emotion dysregulation and NSSI behavior. Extensive research using this measure on a clinical population and the general community demonstrated that people engaged in NSSI have greater difficulty with emotion regulation compared with non-NSSI individuals [44]. In addition, the degree of emotion dysregulation is positively correlated with the frequency of self-injury [45,46]: the greater the degree of emotion dysregulation, the higher the frequency of NSSI behavior. The DERS encompasses six dimensions of emotion regulation in which problems can occur, for example, inability to withhold impulsive behavior when distressed, impaired awareness and understanding of emotion, lack of adaptive regulation strategies to manage negative emotions, difficulties engaging in goal directed behavior when upset, and nonacceptance of negative emotion [38]. Previous research indicated that people with a history of NSSI may have difficulty with emotional awareness [47,48], or a lack of emotion regulation strategies may be correlated with NSSI behavior [49]. Research on the association of impulsivity with NSSI behaviors was mixed [50]. Impulsivity encompasses a wide range of behavioral dimensions, ranging from difficulty maintaining attention on a task and being unconcerned about the consequences of one’s actions, to the pursuit of immediate rewards [51]. Some studies found that difficulty in impulse control is an important emotion regulation deficit for the NSSI group [50], but the measures used in these studies mostly involved self-reporting. In contrast, when laboratory tasks (such as Go/No-Go Task) were applied to measure impulse control, no significant differences between NSSI and non-NSSI groups were found [52]. The differences in the definition and measurement of impulsivity may lead to inconsistent research findings [45]. 

Research showed that adolescents who engage in NSSI behavior have significantly stronger negative emotions such as hostility, guilt, sadness, and anger than non-NSSI individuals [53]. However, a study also indicated that adolescents who display NSSI behavior are less likely to share feelings with family or friends and seem to lack coping resources for dealing with stress or frustration [54]. Thus, they express or regulate their overwhelming but suppressed emotions through NSSI behaviors. For some individuals, the physical pain caused by cutting or skin carving may distract from, or even reduce, emotional pain [25]. 

In conclusion, adolescents who engage in NSSI behaviors may not be a homogeneous group and, therefore, it is essential to learn more about the variation in such behaviors. In this present study, using empirical evidence and a theoretical framework, we firstly investigate whether distinct NSSI subgroups in the adolescent population can be identified according to the frequency of engagement in NSSI. Secondly, we examine the associations of specific dimensions of emotion dysregulation with NSSI subgroups. Much of the previous research on NSSI behavior among adolescents as conducted in developed countries. Thus, this study of Taiwanese female adolescents may enhance our understanding of the complexity and specific characteristics of NSSI in this region. 

## 3. Methods

### 3.1. Recruitment and Participants 

Eleven public high schools in the eastern region of Taiwan were contacted to participate in a larger project conducted by the same authors of the study, entitled “Predictors of NSSI among Adolescents in Taiwan”. The researchers provided and explained assent and consent forms to the students in their classrooms with permission from the school authorities. Initially, a total of 1462 explanatory statement and consent forms were distributed, and 982 students (67.17%) turned in signed consent form on the following day. Of students with parental consent, 809 participants (438 females, 371 males) were present during data collection and completed the questionnaires. The overall participation rate was 55.34%.

Mean age of the total sample was 15.25 *(SD* = 1.20, range = 13–18). The current study included only female participants, whose mean age was 15.23 (*SD* = 1.24, range = 13–18). The mean level of education for mothers and fathers fell between “completed high school/technical diploma” and “college degree”. 

### 3.2. Procedure

Students in Grades 7–12 were administered a set of questionnaires in their classroom. They were assured of anonymity of their responses to the measures. All participants were given a handout listing local mental health services, a 24-hour hotline, and the hospital emergency number. The above research procedure was approved by the Institutional Research Board of the Mennonite Christian Hospital (IRB:11-07-013). 

### 3.3. Measures

The three measures used in this study were validated among an English-speaking population, and Berlin’s [55] back-translation for cross-cultural research procedure was carried out for instrument translation. Firstly, the questionnaire was translated into Chinese by a graduate student who majored in English literature. Secondly, a Taiwanese scholar who studied in the United States for more than 10 years back-translated the Chinese version into English. Thirdly, the researchers in the current study examined and compared the original, the translated, and the back-translated versions. Finally, an evaluation of the equivalence between the measures of the two versions was carried out by recruiting 15 bilingual college students to fill out the questionnaires in both languages. Minor modifications to wording followed to improve clarity for one inconsistent response between the two versions. 

The Difficulties in Emotion Regulation Scale (DERS) [38] is a 36-item measure designed to tap into six dimensions of emotion dysregulation: non-acceptance of negative emotions (e.g., “When I am upset, I feel ashamed at myself for feeling that way”), inability to engage in goal-directed behavior (e.g., “When I am upset, I have difficulty getting work done”), impulse control difficulties (e.g., “When I am upset, I lose control over my behavior”), lack of emotional awareness (e.g., “I care about what I am feeling”), lack of emotional clarity (e.g., “I have no idea how I am feeling”), and limited access to emotion regulation strategies (e.g., “When I am upset, I believe there is nothing I can do to make myself feel better”). Items are rated on a scale of 1 (“almost never, 0–10%”) to 5 (“almost always, 91–100%”), and higher scores indicate more difficulty with regulating emotion. The DERS showed good internal consistency (Cronbach’s α = 0.93) and convergent and construct validity [56,57]. It was also extensively evaluated, having good validity and reliability among adolescents [58,59]. For the present study, internal consistency, assessed via Cronbach’s α, was 0.93 for the total scale, and ranged from 0.76 (non-acceptance) to 0.89 (impulsivity) for subscales. 

Deliberate Self-Harm Inventory (DSHI) [6] is a 17-item self-reported questionnaire that assesses the type and frequency of self-harm behavior (defined as deliberate, direct destruction of body tissue without suicidal intent), including cutting, burning, carving, head-banging, and other NSSI behaviors. Following the suggestion of Gratz, two NSSI scores were created. Firstly, the frequency of the 17 NSSI behaviors was summed to represent the frequency of NSSI. Secondly, the number of NSSI forms employed by participants was calculated by summing the types of NSSI behaviors, which were dummy-coded with “0 = no” or “1 = yes”. Gratz’s study [6] indicated that this scale has high internal consistency (Cronbach’s α = 0.82), satisfactory test–retest (ranging from 2–4 weeks) reliability (ψ = 0.68, *p* < 0.001), and adequate construct and convergent validity. The DSHI demonstrated good reliability among adolescents [60]. Internal consistency for the DSHI in the present sample was high (Cronbach’s α = 0.86). 

The Positive and Negative Affect Schedule (PANAS) was developed by Watson et al. [61] to measure positive and negative affect. The PANAS consists of two 10-item scales that provide brief measures of positive (e.g., “enthusiastic” or “inspired”) and negative (e.g., “distressed” or “upset”) affect. Participants rate their feelings during the past few weeks on a five-point Likert-type scale ranging from 1 (“very slightly or not at all”) to 5 (“extremely”). The score range is 10 to 50, with higher scores on positive affect representing greater intensity of positive emotion and higher scores on negative affect representing greater intensity of negative emotion. The structure of PANAS was validated among adolescents, and the reliability was good for both positive (Cronbach’s α = 0.84) and negative (Cronbach’s α = 0.87) subscales [62]. In this study, reliability was good (positive = 0.86; negative = 0.89).

### 3.4. Data Analysis 

In this study, a two-step cluster analysis was firstly conducted using Ward’s hierarchical clustering method, and the number of NSSIs of the subjects was evaluated by the squared Euclidean distance and dissimilarities, followed by K-means clustering to determine the optimal number of NSSI subgroups. Differences in the self-injury methods of the different NSSI groups were compared by the chi-square test. Analysis of variance (ANOVA) was used to compare differences in emotional regulation difficulties and positive and negative emotions among the different NSSI groups. Bonferroni inequality *t*-test, which was used to counteract inflation of the Type I error rate, was conducted as a follow-up to a significant ANOVA. 

## 4. Results

### Descriptive Statistics 

In total, 161 (36.8%) adolescents reported engaging in at least one NSSI, and, among those reporting NSSI behavior, 77% reported multiple NSSIs during the past year. Intercorrelations (see Table 1) between the research variables showed that NSSI was weakly but significantly correlated with non-acceptance of emotion, difficulties in goal-directed behavior, lack of emotional clarity, and emotional awareness. The magnitude of correlations with NSSI was stronger for lack of strategies and impulse control difficulty than for other dimensions of DERS. In addition, a small negative correlation was obtained between NSSI and positive emotion. Negative affect was moderately correlated with DERS total scores and all DERS subscales except for emotion awareness. There were trends toward small to moderate negative associations between positive affect and DERS total scores and some of the subscales, such as emotion awareness and clarity, emotion regulation strategies, and impulse control.

A two-step cluster analysis was conducted to examine whether the frequency of NSSI behaviors could be used to distinguish subgroups of girls engaging in self-injury. The first group consisted of female adolescents (*n* = 277; aged 13–18 years, mean (M) = 15.38; *SD* = 1.25) who did not carry out self-injurious behaviors. The second group consisted of female adolescents (*n* = 69; aged 13–18 years, M = 15.13; *SD* = 1.15) who previously engaged in moderate self-injurious behavior, with an average age of onset of 13.5 years (*SD* = 1.31) and a mean frequency of NSSI behavior of 4.13 occasions (range 1–8) in the past year; nearly 50% admitted to two or three different methods of NSSI. The third group consisted of female adolescents (*n* = 92; aged 13–18 years, M = 14.83; *SD* = 1.17) who reported severe NSSI behavior, with an average age of onset age of 12.59 years (*SD* = 1.26) and a mean frequency of NSSI behavior of 33.81 occasions (range 9–73) in the past year. Only 4.3% of the female adolescents in this group used just one method of NSSI, while approximately 47% of the group reported using four to six types of NSSI. There were significant differences between the two NSSI groups in terms of age at onset of NSSI (*t* (159) = −4.58, *p* < 0.000, *d* = 0.73), frequency of NSSI behaviors (*t* (159) = 7.83, *p* < 0.000, *d* = 1.25), and number of NSSI types engaged in (*t* (159) = 3.40, *p* < 0.000, *d* = 0.54). As outlined in Table 2, the most common method of NSSI in both groups was cutting, followed by scratching and carving words in the skin. The severe NSSI group was significantly more likely to report more of the following NSSI methods than the moderate group: carving words or pictures into the skin, scratching, biting, and purposely not dealing with wounds, in addition to some other methods.

With respect to the six dimensions of emotion dysregulation and affect, a series of one-way ANOVAs were conducted to test differences in these outcome measures among the NSSI subgroups. Table 3 presents the means and standard deviations of the DERS and the positive and negative affect for the three groups. The results revealed significant differences among the three NSSI groups on scores of six subscales of the DERS, the DERS total score, and both positive and negative affect. The effect sizes of the differences which were indicated by Cohen’s *d* and partial eta squared (*η^2^*) were close to what Cohen [63] described as moderate (e.g., non-acceptance of negative emotions, limited access to emotion regulation strategies, and negative emotion) to large (e.g., impulse control difficulties and DERS total score).

A follow-up Bonferroni inequality *t*-test indicated that participants in the severe NSSI group had significantly higher DERS total, subscale, and negative affect scores than those in the non-NSSI group. Meanwhile, the severe NSSI group reported significantly lower positive affect than the non-NSSI group. The moderate NSSI group also reported significantly higher scores for negative affect, total DERS, non-acceptance of negative emotions, impulse control difficulties, and limited access to emotion regulation strategies than the non-NSSI group. Compared with the moderate NSSI group, female adolescents in the severe NSSI group reported significantly greater negative affect (Cohen’s *d* = 0.53) and difficulty with emotion regulation, as shown by the DERS total score (Cohen’s *d* = 2.31), lack of emotion regulation strategies (Cohen’s *d* = 0.66), and difficulty in impulse control (Cohen’s *d* = 0.63). 

## 5. Discussion 

In the present study, we investigated whether distinct NSSI subgroups of adolescents can be identified based on differences in the number of NSSI behaviors in which they engage. In addition, the relationships between specific dimensions of emotion dysregulation and NSSI subgroups were further explored. Several interesting findings emerged from this study.

### 5.1. Overview of NSSI Behaviors of Female Adolescents

Approximately 37% of the participants reported NSSI behavior at least once during the past year, consistent with previous research conducted on nonclinical samples of adolescents [13,20,64]. However, this prevalence rate is higher than that reported in previous school-based studies of adolescents [15,17]. The variability in the range of prevalence may be due to differences in the inventories used and the time frame over which NSSI was assessed. For example, in Wang et al.’s [14] study, NSSI behaviors were assessed over the past six months, but the time period covered by the assessment of NSSI in the current study was 12 months. There were 17 NSSI behaviors assessed in our study; however, in Tang et al.’s [15] school-based study, they only evaluated 10 NSSI behaviors. Despite these differences, our findings indicate that the onset and maintenance of NSSI behaviors are significant problems for adolescents.

Of the female adolescents who participated in this study, 21% admitted to engaging in more than nine NSSIs, and we identified this subgroup as the severe NSSI group. We speculate that their NSSI behavior might not have occurred out of curiosity or occasional experimentation, as suggested by the results of Klonsky and Olino [34]. The moderate NSSI group represented 16% of the sample, and the girls in this group reported an average of four NSSI engagement during the past year. Female adolescents in both groups reported an age of onset of NSSI behavior between 12 and 14, which is in line with previous studies [22,25,26,27,65]. It is worth mentioning that the reported age of onset of NSSI was nearly one year younger in the severe group compared with the moderate group. In both groups, the commonly reported methods of NSSI were cutting, scratching, and carving words into the skin. The results of our study showed that half of the girls in the severe group engaged in NSSI by intentionally preventing wounds from healing, which might be overlooked by parents or teachers as an NSSI behavior. The findings suggest that administration of NSSI screening instruments (such as DSHI) to students in general can be a preliminary step, which is expected to identify the potential presence of NSSI behaviors; a subsequent thorough assessment should proceed based on the screening results. 

### 5.2. Association of Dimensions of Emotion Dysregulation and NSSI Subgroups

In line with our expectations, teenage girls who engaged in NSSI behavior, whether in the severe or moderate group, had higher negative affect than teenage girls who exhibited no NSSI engagement. When experiencing negative emotions, girls in both NSSI groups may have greater difficulty with acceptance of emotion, access to emotion regulation strategies, and control of impulsive behaviors. In addition, female adolescents in the severe NSSI group showed even higher negative affect and more difficulty with acceptance of negative emotion and impulse control than those in the moderate group. These findings echo those of previous studies [12,27] and indicate that a high level of negative emotion and lack of emotion regulation are important features of NSSI. Individuals who engage in NSSI usually have an obviously lower ability to recognize, understand, process, and express emotions than those who do not [45]. Salters-Pedneault et al. [66] suggested that, when individuals are unable to adaptively regulate negative emotion but choose to suppress or avoid the feeling, it may gradually accumulate and eventually rebound, causing the individuals to experience more intense negative emotions.

The identification of emotion dysregulation as a robust predictor for NSSI is consistent with the results of other studies [29,48]. Adolescents who repetitively harm themselves are usually motivated to reduce their negative emotions through cutting or other NSSI behaviors [32], which serve as fast or convenient methods to alleviate negative emotion for adolescents whose emotion regulation skills are limited [7,12,26,27,36]. Kimball and Diddams [67] also reported that youths possessing the capability to regulate negative emotion are less likely to engage in behaviors causing harm to their bodies than those who lack this capability. In our study, female adolescents who engaged in either severe or moderate NSSI behaviors usually had lower tolerance to their own negative emotions and often performed acts without thinking about the consequences or other possibilities. Combining the findings of this study with those of other studies, it can be inferred that teenagers with a limited emotion regulation capacity who are unable to accept their painful emotions are more likely to use NSSI to regulate their negative emotions. NSSI relieves negative emotions but leads to feelings of guilt and shame about one’s own behaviors. Additionally, self-injurious behaviors only briefly regulate an individual’s emotion in the moment and are not capable of relieving the pressures that the individual suffers. To deal with these multiple painful feelings, NSSI may once again be adopted, leading to a vicious long-term cycle [25,42].

The results of this study show that difficulty with impulse control is an important predictor of self-injurious behavior, and the severe NSSI group had significantly greater difficulty in impulse control than the moderate NSSI group. Such results are consistent with those of studies that measured impulse control with a self-reporting scale [68,69]. For self-injurious behavior, the process of intention to action may take only a few minutes; therefore, a lack of self-control and limited strategies for emotion regulation may lead to self-injurious behaviors under the drive of impulse. Many self-injured people described that impulsive behavior, whether cutting, scratching, or carving pictures into the skin, can relieve emotional pain or shift it to a physical pain, thus negatively reinforcing the NSSI behavior [25,51,70].

### 5.3. Clinical Implications 

The results of this study provide some evidence of the universality and severity of NSSI behavior among female adolescents. The high rate of NSSI behavior underscores the importance of examining the frequency and severity of NSSI, as well as the associated risk factors such as capability of regulating negative emotion, in assessments of adolescent mental health. Adolescents who engage in NSSI multiple times and demonstrate overall emotion dysregulation may benefit from evidence-based treatments such as cognitive behavioral therapy (CBT), acceptance and commitment therapy (ACT), or dialectical behavioral therapy (DBT). Both CBT and DBT involve the development of the components of building emotional awareness, as well as skills to regulate emotion. Significant effects of CBT on the development of problem-solving by adolescents were also demonstrated [71]. Mindfulness-based therapies such as ACT and DBT can be used to teach teenagers to accept emotions in the moment, whether positive or negative, without resisting or exaggerating the emotion [72]. Importantly, DBT also teaches adolescents with impulse control problems to analyze chains of maladaptive behaviors and manage contingency actions, such as punishment or reinforcement [73]. 

### 5.4. Limitations and Future Directions 

The present study contributes to the understanding of the relationship between specific emotion dysregulation and NSSI behavior in female adolescents. However, it is limited by its cross-sectional design and sole reliance on retrospective reports in the assessment. Recall bias may affect adolescents’ reports of NSSI behavior and emotion regulation, and only correlational conclusions can be drawn from the current results. Future studies could examine the antecedents and consequences of NSSI by adopting the experience sampling method [74], which involves the assessment of adolescents’ behaviors and emotions multiple times a day across a certain time period. These data can provide a more comprehensive picture of the fluctuation of emotions and its relationship with NSSI. Secondly, although we recruited participants across multiple sites to account for variations that may exist among different schools, the sample was relatively homogeneous in terms of demographic and background characteristics. Future research could consider a wider geographic area and include treatment-seeking adolescents both at school and in clinical settings. Finally, some important variables that may be related to NSSI were not specifically assessed such as social support, history of childhood abuse, clinical history of diagnosis, and NSSI behavior of family members or close friends. These variables could be included in future research to investigate the etiology, maintenance, or course of NSSI. 

## 6. Conclusions

Despite these limitations, the present study provides important information about the prevalence of NSSI in a school-based sample in Taiwan and the distinct features of NSSI subgroups of female adolescents. The findings indicate that self-injurious female adolescents are more likely to experience higher negative affect and show lower emotional regulation ability, including acceptance of emotions and impulse control. The results suggest that assessments of NSSI and emotion regulation should be incorporated in youth mental health screening. 

## Figures and Tables

**Table 1 ijerph-16-03361-t001:** Correlations between non-suicidal self-injury (NSSI) and other research variables (*n* = 438)

Variables	1	2	3	4	5	6	7	8	9	10
DERS										
1. Strategies	-									
2. Nonacceptance	0.66 **	-								
3. Impulse	0.72 **	0.59 **	-							
4. Goals	0.60 **	0.49 **	0.56 **	-						
5. Awareness	0.18 **	0.07	0.17 **	0.02	-					
6. Clarity	0.40 **	0.32 **	0.36 **	0.17 **	0.66 **	-				
7. DERS total	0.87 **	0.75 **	0.83 **	0.68 **	0.45 **	0.64 **	-			
8. Positive affect	−0.24 **	−0.04	−0.16 **	−0.08	−0.43 **	−0.29 **	−29 **	-		
9. Negative affect	0.56 **	0.48 **	0.53 **	0.36 **	0.04	0.24 **	0.54 **	0.10 *	-	
10. NSSI	0.33 **	0.27 **	0.39 **	0.14 **	0.22 **	0.24 **	0.38 **	−0.13 **	0.31 **	-

DERS: Difficulties in Emotion Regulation Scale. * *p* < 0.05; ** *p* < 0.01.

**Table 2 ijerph-16-03361-t002:** Frequency and percentage of Deliberate Self-Harm Inventory (DSHI) items engaged in by the two non-suicidal self-injury groups (*n* = 161).

	Self-Harm Behavior	Moderate NSSI *n* = 69	Severe NSSI *n* = 92	χ^2^ (1)	Cramer’s V
1.	Cut wrist, arms, or other area(s) of body	57 (82.61)	77 (83.70)	0.03	0.01
2.	Burned with a cigarette	1 (1.44)	20 (21.73)	14.31 **	0.30
3.	Burned with a lighter or a match	1 (1.44)	7 (7.60)	3.17	0.14
4.	Carved words into skin	16 (23.19)	49(53.26)	14.81 **	0.30
5.	Carved pictures, designs, or other marks into skin	9 (13.04)	40 (43.47)	17.25 ***	0.33
6.	Severely scratched resulting in scarring or bleeding	16 (23.19)	63 (68.48)	32.36 ***	0.45
7.	Bit to the extent of breaking the skin	13 (18.84)	41 (44.57)	11.71 ***	0.27
8.	Dripped acid onto skin	0 (0)	2 (2.17)	1.52	0.10
9.	Rubbed sandpaper on the body	0 (0)	7 (7.60)	5.49 *	0.19
10.	Used bleach, comet, or oven cleaner to scrub skin	0 (0)	3 (3.26)	2.29	0.12
11.	Stuck sharp objects such as needles, pins, and staples into skin	4 (5.80)	29 (31.52)	16.01 ***	0.32
12.	Rubbed glass into skin	3 (4.35)	30 (32.61)	19.33 ***	0.35
13.	Broken bones	0 (0)	9 (9.78)	7.15 *	0.21
14.	Banged head against something to cause a bruise	3 (4.35)	21 (22.82)	10.61 **	0.26
15.	Punched to cause a bruise to appear	3 (4.35)	29 (31.52)	18.28 ***	0.34
16.	Prevented wounds from healing	8 (11.59)	46 (50.0)	26.09 ***	0.40

* *p* < 0.05, ** *p* < 0.01, *** *p* < 0.001.

**Table 3 ijerph-16-03361-t003:** Comparison of the means of variables of interest between groups.

Variables	Non-NSSI (*n* = 277) M (SD)	Moderate NSSI (*n* = 69) M (SD)	Severe NSSI (*n* = 92) M (SD)	Group Comparison F(2435) *d* *(η^2^)*	Post Hoc Comparison
DERS strategies	19.18(6.73)	21.43(7.10)	25.60(6.95)	30.74 *** 0.68 (0.12)	s > m > n
DERS non-acceptance	15.08(4.73)	16.80(4.85)	18.53(5.09)	18.56 *** 0.56 (0.08)	s > n, m > n
DERS impulse	12.73(5.18)	14.94(6.13)	18.92(6.28)	42.91 *** 0.80 (0.17)	s > m > n
DERS goals	15.94(4.85)	16.30(4.99)	17.99(4.69)	6.23 ** 0.27 (0.03)	s > n
DERS awareness	14.40(4.69)	15.53(5.27)	17.24(5.46)	11.56 *** 0.43 (0.05)	s > n
DERS clarity	10.29(3.64)	11.47(4.24)	12.69(4.56)	13.37 *** 0.48 (0.06)	s > n
DERS total	87.60(20.72)	96.46(24.35)	110.97(21.38)	41.44 *** 0.79 (0.16)	s > m > n
Negative affect	22.04(7.94)	25.45(9.08)	28.83(10.24)	22.44 *** 0.62 (0.09)	s > m > n
Positive affect	29.87(7.58)	28.58(8.60)	27.36(8.70)	3.60 * 0.25 (0.02)	s > n

DERS: Difficulties in Emotion Regulation Scale, NSSI: non-suicidal self-injury, M: mean, SD: standard deviation, *d*: effect size, *η^2^:* partial eta squared; * *p* < 0.05, ** *p* < 0.01, *** *p* < 0.001.

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
