# Peer review of "Association between Emotion Dysregulation and Distinct Groups of Non-Suicidal Self-Injury in Taiwanese Female Adolescents"

_ijerph, 2019, doi:10.3390/ijerph16183361_

Round 1

Reviewer 1 Report

Overall, this is an interesting and important paper to the field. The paper can be improved. The following are suggestions for improvement:

1. Abstract- add Taiwan
2. Background- need to add a background about NSSI among Taiwanese female adolescents (e.g. what do we know such as NSSI female adolescents, gender differences)?
3. Methods- reorganize Methods section to 1) Overall/Summary paragraph of methods/procedures/recruitment; 2) Participants' inclusion/exclusion criteria; 3) Measures; 4) Data analysis

a. Line 133- suggestion to change "conducted" with "validated among English-speaking population"
b. Lines 140-141- Were there a lot of word modifications? This sentence needs more description
c. Line 171- what does "group-administered" mean?
d. Lines 198-206- The categorization of groups is unclear. Is it "ever" or "lifetime" or just previous year? Sometimes, there's previous "years", but is this "lifetime" or "X number of previous years"? Also, previous year (singular) is just past 12 months.

5. Results- Write out NSSI in the title of Table 2

6. Discussion- needs subtitles as it is hard to follow/read
Lines- 261-262- What's the recommended protocol for NSSI assessment in Taiwan and/or globally? What is the current protocol for NSSI assessment now, if different?

Author Response

Thank you for your very careful review of our manuscripts, and for the comments, corrections and suggestions that ensued. Your comments helped us refine our paper by clarifying critical sections and presenting results. Please refer to the manuscript for numerous changes motivated by your comments. Below we summarize major comments you provided and our reaction to them.

Point 1 Abstract- add Taiwan

Response 1: Thanks for the suggestion. Modified as suggested. (Line 16)

Point 2 Background- need to add a background about NSSI among Taiwanese female adolescents (e.g. what do we know such as NSSI female adolescents, gender differences)?

Response 2: Thanks for the suggestion. In the revision, we provided information of prevalence of NSSI behavior in Taiwan as well as gender differences.

Past research also indicated high prevalence of NSSI in Taiwan. Lin et al. [26] reported a prevalence of 20.1% among Taiwanese high school students, and females reported significantly higher frequency of NSSI than did males. (Line 64-66)

Point 3 Methods- reorganize Methods section to 1) Overall/Summary paragraph of methods/procedures/recruitment; 2) Participants' inclusion/exclusion criteria; 3) Measures; 4) Data analysis

Response 3: Thanks for pointing this out. We re-organization Methods section as suggested. (1) Recruitment and participants; (2) Procedure; (3)Measures; (4) Data analysis. (Line 133-205)

Point 4 Line 133- suggestion to change "conducted" with "validated among English-speaking population"

Response 4: Thanks for pointing this out. In the revision, the sentence has been revised as suggested. ( Line 154)

Point 5 Lines 140-141- Were there a lot of word modifications? This sentence needs more description

Response 5: Thanks for pointing this out. Indeed, the sentence can be misleading. Only minor word modification was made in Chinese version. We have revised the sentences and made the content clearer.

Minor modifications to wording followed to improve clarity for one inconsistent response between the two versions. (Line 161-162)

Point 6 c. Line 171- what does "group-administered" mean?

Response 6: Thanks for pointing this out. We have revised the sentences and made the content clearer.

Students in Grades 7-12 were administered a set of questionnaires in their classroom. They were assured of anonymity of their responses to the measures. (Line 148-152)

Point 7 d. Lines 198-206- The categorization of groups is unclear. Is it "ever" or "lifetime" or just previous year? Sometimes, there's previous "years", but is this "lifetime" or "X number of previous years"? Also, previous year (singular) is just past 12 months.

Response 7: Thanks for pointing this out. Indeed, the sentence can be misleading. In the revision, we have replaced “previous year ” by “the past year” (Line 209, 228, 231, and 289)

Point 8 Results- Write out NSSI in the title of Table 2

Response 8: Thank you for pointing this out. Modified as suggested. (Line 252)

Point 9 Discussion- needs subtitles as it is hard to follow/read

Response 9: Thank you for the suggestion. We added four subheadings in the section of discussion. They are: 1. Overview of NSSI behaviors of female adolescents; 2. Association of dimensions of emotion dysregulation and NSSI subgroups; 3. Clinical implicaiotns; 4. Limitations and future directions. (Line 273, 301, 342, and 358)

Point 10 Lines- 261-262- What's the recommended protocol for NSSI assessment in Taiwan and/or globally? What is the current protocol for NSSI assessment now, if different?

Response 10: Thanks for the very thoughtful comment. NSSI behavior is not included in the mental health problems screen in Taiwan. In the revision, we provided more detailed information of protocol for NSSI assessment.

The findings suggest that administration of NSSI screening instruments (such as DSHI) to students in general can be a preliminary step, which is expected to identify the potential presence of NSSI behaviors; a subsequent thorough assessment should proceed based on the screening results. (Line 296-299)

Reviewer 2 Report

Introduction 

Overall, the Introduction is well written and well organized. It does an excellent job of establishing the significance of the study while also connecting it to past theoretical and empirical literature related to NSSI, emotion regulation, development, and psychosocial functioning. A few minor changes, however, could strengthen and/or clarify this section, as detailed below:

The authors might consider expanding the definition of NSSI and specifically, further clarifying what hostile means or re-wording this aspect of the definition. Hostile can evoke a connotation of NSSI as being inherently intense. NSSI can be very intense and violent, but episodes of self-injury can also be mild. 

Regarding line 90, Negative reinforcement entails the increased likelihood versus automatic certainty of an operant occurring in the future. Thus, the authors should consider tempering this sentence (e.g., …implying that NSSI may become more likely and/or frequent, particularly in similar contexts). 

As written now, the section on impulsivity (i.e., lines 101–109) seems somewhat distinct from the rest of the content reviewed in the Introduction. The authors might re-organize the sentences about impulsivity to more strongly identify how impulsivity is relevant to emotion regulation in the context of NSSI and incorporate this into the flow of this content area

Methods

Overall, the Methods used are appropriate for the study. However, several changes are needed, as detailed below.

The authors helpfully provide the sample’s mean age, but the range of ages (overall and per subsample) would also be useful, given that age (as the authors note) is related to onset and gender differences. The authors provide the age range for the overall sample in the abstract, but not in in-text (and ranges for subsamples do not seem to be provided anywhere).

The authors indicate the 438 girls were part of a larger sample of 809 youth. This seems to suggest 371 adolescent boys also completed the measures. The authors might consider reporting their results as well to see the degree to which NSSI levels and associations with emotion dysregulation generalize or vary significantly across genders. Alternatively, the authors might consider preparing a secondary manuscript which tests for gender differences with this sample.  

On line 133, the authors state the measures were conducted in English. However, they then go on to describe how the measures were translated (quite rigorously too). Perhaps Line 133 contains a typographical error, as it seems the measures–which were originally in English–were translated into Chinese (presumably Taiwanese Mandarin?) and then administered to the sample in their Chinese versus English forms. Clarification by the authors would be helpful.  

Regarding measures, the authors describe the DERS’ six dimensions and provide example items in parentheses. However, since they only provide one per dimension, they should preface each parenthetical example with “e.g.”. For all three measures, the authors should state explicitly whether each measure has been validated with adolescents of the same age range as their sample, or at least the next-closest age range with which each measure has been validated. Finally, the authors report the internal consistency of the DERS, DSHI, and PANAS with prior samples (which is important), but they should also include the DERS’, DSHI’s, and PANAS’ internal consistency with the present sample (which is more important). 

If possible, the authors should report the approximate response rate, as the number of respondents is known and the number of students who were recruited might also be known and/or easily estimated. Prior studies have shown that response rates to NSSI-related constructs such as traumatic stress can bias reported rates of studied variables (see Lines 247–248). 

Results

When reporting the correlational results in-text (versus in the table), the authors should recognize the relative magnitude of the correlations (i.e., correlations of .10–.29 are typically categorized as small, .30-.49 are moderate or medium, and .50+ are large). For example, NSSI was positively related to all DERS’ dimensions to a statistically significant degree. However, the magnitude of association varied notably, with moderate/medium relations between NSSI and DERS-strategies (.33) and -impulse (.39), but only small relations between NSSI and DERS-goals (.14) and -awareness (.22). 

When reporting t-test results (e.g., Lines 209–210), the authors should also report effect sizes (i.e., Cohen’s d). 

The use of a Bonferroni correction, while appropriate, should be explained in the Data Analysis section (lines 178–184) rather than first mentioned in the Results section. 

For the chi-squared tests, the authors need to explicitly list an effect size (e.g., phi, Cramer’s V, or odds ratio) for each test (significant and otherwise).

For ANOVAs’ main-effects, the traditional effect size is partial eta squared; whereas, follow-up contrasts use Cohen’s d. Both should be reported, if not in-text, then at least in Table 3.

Discussion

Overall, the Discussion section is well organized and well written and ably interprets the authors' findings, particularly in the context of past empirical and theoretical literature. One exception is when the authors note they found a higher prevalence rate than found by other studies; they mention that this higher rate might be due to differences in measured used and time-frames of assessment. The authors should expand this section to more explicitly explain why and/or how their measures (as well as time-frames) compared to others might have biased rates. The authors might also wish to consider that differences in rates might not have been due to measurement error, but were veridical, and perhaps consider what factors might explain the differences (e.g., certain cultural elements). 

General 

Overall, the manuscript is well written; however, there are a few typographical and/or syntactical errors that could be easily remedied, as noted below:

Line 16: The study evaluated or examined the hypothesized association, versus “sought to associate...” or make variables become correlated. Please revise accordingly.

Line 45: There appears to be an extraneous space between by and suicidal.

Line 62. Authors might change have to endorse or report, since the cited studies used self-report versus observational or physiological measures of psychological distress.

Line 94: The authors refer to both clinical and the general population; please revise to say both clinical and general community samples (as the cited studies used samples, not population data).  

Line 165: Suggest editing the scale response descriptions to match the same format used for describing the DERS response options (i.e., 1 [very slightly or not at all]).

Lines 194 and 222: N should be italicized (i.e., = 438).

Lines 235–239: The sentence is cut off and then erroneously added into Table 3.

Line 236: Means should not be italicized. 

Line 236: ES should be listed as (as there are multiple types of effect sizes, and the common abbreviation of Cohen’s in tables is simply d). 

Line 255: There appears to be an extraneous space between an and age.

Line 290: The sentence ends with a preposition (i.e., from), which is grammatically incorrect.

Lines 306–307: Per APA style rules, the names of these therapies should not be capitalized.

Line 309: There should be a hyphen between problem and solving (i.e., problem-solving). 

Line 319: Per APA style rules, experience sampling method (ESM) should not be capitalized. 

Author Response

Thank you for your very careful review of our manuscripts, and for the comments, corrections and suggestions that ensued. Your comments helped us refine our paper by clarifying critical sections and presenting results. Please refer to the manuscript for numerous changes motivated by your comments. Below we summarize major comments you provided and our reaction to them.

Point 1 The authors might consider expanding the definition of NSSI and specifically, further clarifying what hostile means or re-wording this aspect of the definition. Hostile can evoke a connotation of NSSI as being inherently intense. NSSI can be very intense and violent, but episodes of self-injury can also be mild. 

Response 1: Thanks for pointing this out. In the revision, the sentence has been revised as suggested.

NSSI comprises behaviors in which people intentionally inflict damage to the surface of their body [4] (line 42)

Point 2 Regarding line 90, Negative reinforcement entails the increased likelihood versus automatic certainty of an operant occurring in the future. Thus, the authors should consider tempering this sentence (e.g., …implying that NSSI may become more likely and/or frequent, particularly in similar contexts). 

Response 2: Thanks for the very thoughtful suggestions. We have revised the relevant section and made the content clearer. 

implying that individuals adopt NSSI as coping strategy for regulating aversive emotional experiences. The positive correlation between the frequency of NSSI and the function of negative reinforcement was indicated by Klonsky [41]. (Line 94-96)

Point 3 As written now, the section on impulsivity (i.e., lines 101–109) seems somewhat distinct from the rest of the content reviewed in the Introduction. The authors might re-organize the sentences about impulsivity to more strongly identify how impulsivity is relevant to emotion regulation in the context of NSSI and incorporate this into the flow of this content area.

Response 3: Thank you for the very thoughtful comment. We have revised the relevant section and made the content clearer. 

 The DERS encompasses six dimensions of emotion regulation in which problems can occur; for example, inability to withhold impulsive behavior when distressed, impaired awareness and understanding of emotion, lack of adaptive regulation strategies to manage negative emotions, difficulties engaging in goal directed behavior when upset, nonacceptance of negative emotion [38].(Line 103-107).

Point 4 The authors helpfully provide the sample’s mean age, but the range of ages (overall and per subsample) would also be useful, given that age (as the authors note) is related to onset and gender differences. The authors provide the age range for the overall sample in the abstract, but not in in-text (and ranges for subsamples do not seem to be provided anywhere).

Response 4: Thanks for the suggestion. We have added mean age and age range for overall sample and three subgroups. (Line 145-146, 228-229, and 233)

Point 5 The authors indicate the 438 girls were part of a larger sample of 809 youth. This seems to suggest 371 adolescent boys also completed the measures. The authors might consider reporting their results as well to see the degree to which NSSI levels and associations with emotion dysregulation generalize or vary significantly across genders. Alternatively, the authors might consider preparing a secondary manuscript which tests for gender differences with this sample.  

Response 5: Thanks for the very thoughtful suggestions. Actually, the study which examined the NSSI behaviors of the 371 male adolescents was published by the same authors in 2017. The reference is listed as below.

Chun, CC, Chen, WL. Attachment style, emotion dysregulation, and non-suicidal self-harm behavior of male adolescents: the mediating role of negative emotion and emotion dysregulation. Chang Gung J. Humanities and Social Science. 2017; 10(1): 1-41.     

Point 6 On line 133, the authors state the measures were conducted in English. However, they then go on to describe how the measures were translated (quite rigorously too). Perhaps Line 133 contains a typographical error, as it seems the measures–which were originally in English–were translated into Chinese (presumably Taiwanese Mandarin?) and then administered to the sample in their Chinese versus English forms. Clarification by the authors would be helpful.  

Response 6: Thank you for pointing this out. Indeed, it’s a typographical error.  In the revision, the sentence has been revised as suggested. (Line 157)

Point 7 (a)Regarding measures, the authors describe the DERS’ six dimensions and provide example items in parentheses. However, since they only provide one per dimension, they should preface each parenthetical example with “e.g.”.

(b) For all three measures, the authors should state explicitly whether each measure has been validated with adolescents of the same age range as their sample, or at least the next-closest age range with which each measure has been validated.

(c) Finally, the authors report the internal consistency of the DERS, DSHI, and PANAS with prior samples (which is important), but they should also include the DERS’, DSHI’s, and PANAS’ internal consistency with the present sample (which is more important). 

Response 7: (a) Thanks for pointing that out. Modified as suggested. (Line 168-173, and Line 193).

(b) Thanks for the suggestions. We have added psychometric properties of the measures which were tested among adolescents. (Line 177-179, Line 188-190, and Line 197-200)

(c) Thanks for the suggestions. We have added internal consistency of the measures with the present sample. (Line 178-179, Line 189-190, and Line 199-200)

 Point 8 If possible, the authors should report the approximate response rate, as the number of respondents is known and the number of students who were recruited might also be known and/or easily estimated. Prior studies have shown that response rates to NSSI-related constructs such as traumatic stress can bias reported rates of studied variables (see Lines 247–248). 

Response 8: Thanks for the suggestions. We have revised the relevant section and made the content clearer.

Initially a total of 1462 explanatory statement and consent forms were distributed, and 982 students (67.17%) turned in signed consent form on the following day. Of students with parental consent, 809 participants (438 females, 371 males) were present during data collection and completed questionnaires. The overall participation rate was 55.34%. (Line 140-144)

Point 9  When reporting the correlational results in-text (versus in the table), the authors should recognize the relative magnitude of the correlations (i.e., correlations of .10–.29 are typically categorized as small, .30-.49 are moderate or medium, and .50+ are large). For example, NSSI was positively related to all DERS’ dimensions to a statistically significant degree. However, the magnitude of association varied notably, with moderate/medium relations between NSSI and DERS-strategies (.33) and -impulse (.39), but only small relations between NSSI and DERS-goals (.14) and -awareness (.22). 

Response 9:  Thanks for pointing this out. We have revised the paragraph as suggested.

NSSI was weakly but significantly correlated with non-acceptance of emotion, difficulties in goal-directed behavior, lack of emotional clarity, and emotional awareness. The magnitude of correlations with NSSI was stronger for lack of strategies and impulse control difficulty than for other dimensions of DERS. In addition, a small negative correlation was obtained between NSSI and positive emotion. Negative affect was moderately correlated with DERS total scores and all DERS subscales except for emotion awareness. There were trends toward small to moderate negative associations between positive affect and DERS total scores and some of the subscales, such as emotion awareness and clarity, emotion regulation strategies, and impulse control. (Line 214-222)

Point 10  When reporting t-test results (e.g., Lines 209–210), the authors should also report effect sizes (i.e., Cohen’s d). 

Response 10: Thanks for pointing this out. We added effect size for each t test. 

…at onset of NSSI (t (159) = –4.58, p < 0.000, d = 0.73), frequency of NSSI behaviors (t (159) = 7.83, p < 0.000, d = 1.25), and number of NSSI types engaged in (t (159) = 3.40, p < 0.000, d = 0.54). (Line 238-240)

Point 11 The use of a Bonferroni correction, while appropriate, should be explained in the Data Analysis section (lines 178–184) rather than first mentioned in the Results section. 

Response 11: Thanks for the comments. We have revised the relevant section and made the content clearer. 

Bonferroni inequality t test, which was used to counteract inflation of the Type I error rate, was conducted follow-up to a significant ANOVA. (Line 208-209)

Point 12 For the chi-squared tests, the authors need to explicitly list an effect size (e.g., phi, Cramer’s V, or odds ratio) for each test (significant and otherwise).

Reponse 12: Thanks for pointing this out. We added Cramer’s V for each chi square analysis in Table 2. 

Point 13 For ANOVAs’ main-effects, the traditional effect size is partial eta squared; whereas, follow-up contrasts use Cohen’s d. Both should be reported, if not in-text, then at least in Table 3.

Reponse 13: Thanks for pointing this out. We added partial eta squared as effect size for each outcome variables in Table 3. (Line 268 , Table 3)  

Point 14 One exception is when the authors note they found a higher prevalence rate than found by other studies; they mention that this higher rate might be due to differences in measured used and time-frames of assessment. The authors should expand this section to more explicitly explain why and/or how their measures (as well as time-frames) compared to others might have biased rates. The authors might also wish to consider that differences in rates might not have been due to measurement error, but were veridical, and perhaps consider what factors might explain the differences (e.g., certain cultural elements). 

Reponse 14: Thanks for the very thoughtful suggestions. We have revised the relevant section and made the content clearer. 

For example, in Wang et. al.’s [14] study, NSSI behaviors were assessed over the past 6 months, but the time period covered by the assessment of NSSI in the current was 12 months. There were 17 NSSI behaviors being assessed in our study, however in Tang et al.’s [15] school-based study, they only evaluated 10 NSSI behaviors. Despite these differences, our findings indicate that the onset and maintenance of NSSI behaviors are significant problems for adolescents. ( Line 283-288)

Point15 Line 16: The study evaluated or examined the hypothesized association, versus “sought to associate...” or make variables become correlated. Please revise accordingly.

Reponse 15: Thanks for the suggestion. Modified as suggested. (Line 16)

Point 16 Line 45: There appears to be an extraneous space between by and suicidal.

Reponse 16: Thanks for pointing this out. Modified as suggested. (Line 45)

Point 17 Line 62. Authors might change have to endorse or report, since the cited studies used self-report versus observational or physiological measures of psychological distress.

Reponse 17: Thanks for the suggestion. Modified as suggested.

Many self-injured people reported to use multiple…(Line 68)

Point 18  Line 94: The authors refer to both clinical and the general population; please revise to say both clinical and general community samples (as the cited studies used samples, not population data).  

Reponse 18: Thanks for the suggestion. Modified as suggested. (Line 99)

Point 19 Line 165: Suggest editing the scale response descriptions to match the same format used for describing the DERS response options (i.e., 1 [very slightly or not at all]).

Reponse 19: Thanks for the suggestion. Modified as suggested. (Line 195)

Point 20 Lines 194 and 222: N should be italicized (i.e., = 438).

Reponse 20: Thanks for pointing this out. Modified as suggested. (table 1, Line 223, table 2, Line 256 )

Point 21 Lines 235–239: The sentence is cut off and then erroneously added into Table 3.

Reponse 21: Thanks for pointing this out. Modified as suggested.

Point 22 Line 236: Means should not be italicized. 

Reponse 22: Thanks for pointing this out. Modified as suggested.

Point 23 Line 236: ES should be listed as (as there are multiple types of effect sizes, and the common abbreviation of Cohen’s in tables is simply d). 

Reponse 23: Thanks for pointing this out. Modified as suggested. (Table 3)

Point 24 Line 255: There appears to be an extraneous space between an and age.

Reponse 24: Thanks for pointing this out. Modified as suggested.

Point 25 Line 290: The sentence ends with a preposition (i.e., from), which is grammatically incorrect.

Reponse 25: Thanks for the suggestion. Modified as suggested.

Point 26 Lines 306–307: Per APA style rules, the names of these therapies should not be capitalized.

Reponse 26: Thanks for the suggestion. Modified as suggested. (Line 353-354)

Point 27  Line 309: There should be a hyphen between problem and solving (i.e., problem-solving). 

Reponse 27: Thanks for the suggestion. Modified as suggested. (Line 356)

Point 28 Line 319: Per APA style rules, experience sampling method (ESM) should not be capitalized. 

Reponse 28: Thanks for the suggestion. Modified as suggested. (Line 370)